# Peste des Petits Ruminants Vaccine: Criteria for Assessing Its Thermotolerance

**DOI:** 10.3390/v17091151

**Published:** 2025-08-22

**Authors:** Charles S. Bodjo, Hassen Belay Gelaw, Zione D. Luhanga, Yebechaye Degefa Tessema, Jean-De-Dieu Baziki, Cisse R. Moustapha Boukary, Gelagay Ayelet Melesse, Ethel Chitsungo, Nick Nwankpa, Simon Kihu, Felix Njeumi, Satya Parida, Adama Diallo

**Affiliations:** 1African Union-Pan African Veterinary Vaccine Centre (AU-PANVAC), Debre Zeit P.O. Box 1746, Ethiopia; hasseng@africanunion.org (H.B.G.); zioneluhanga0@gmail.com (Z.D.L.);; 2World Organisation for Animal Health (WOAH), Sub-Regional Representation for Eastern Africa, Nairobi P.O. Box 19687, Kenya; s.kihu@woah.org; 3Food and Agriculture Organization of the United Nations (FAO), Viale delle Terme di Caracalla, 00153 Rome, Italy; felix.njeumi@fao.org (F.N.); satya.parida@fao.org (S.P.); 4Independent Researcher, Hahngasse 24-26/02 07, 1090 Vienna, Austria; a.diallob@outlook.com

**Keywords:** PPR vaccine, titre, thermotolerance, criteria for the thermotolerance, titre loss

## Abstract

The Peste des Petits Ruminants (PPR) live attenuated vaccines, the PPR virus (PPRV) Nigeria 75/1 strain (lineage II) and PPRV India Sungry 96 strain (lineage IV), currently used for control and eradication programme are very efficient vaccines as they provide the host, sheep and goats, a lifelong immunity after a single minimum recommended dose of 10^2.5^ TCID_50_/mL. Unfortunately, both live attenuated vaccines are thermolabile and their use requires maintaining the cold chain from the manufactory premises to the field as most PPR-infected regions are facing of hot climate, with poor infrastructure, and the maintenance of an effective cold chain remains a challenge. To address this challenge, efforts have focused on developing thermotolerant (ThT) PPR vaccines using different stabilisers and improving the freeze-drying process. This study aimed to define the criteria for the evaluation of the stability of ThT PPR vaccines. A total of 37 batches of freeze-dried PPR vaccines using the PPRV Nigeria 75/1 strain, including eight (8) and twenty-nine (29) vaccines labelled as ThT and conventional formulations, respectively, were tested to evaluate the stability at temperatures of 40 °C to simulate the field conditions in some hot climate regions. All the vaccine batches included in this study initially showed acceptable levels of residual moisture, below 3%, and titres above the minimum WOAH standard requirement of 10^2.5^ TCID_50_/mL. Following the incubation at 40 °C, 56.7% and 46% of the 37 vaccine batches tested retained titres above 10^2.5^ TCID_50_/mL on day 3 and day 5, respectively. These vaccines use stabilisers such as skimmed milk, lactalbumin–sucrose, trehalose and one unnamed product (which may be protected for patent). The mean of titre loss among the PPR vaccines maintaining titres above 10^2.5^ TCID_50_/mL was 0.78 log10 at day 3 and 0.99 log10 at day 5, suggesting a significant early degradation during the first 3 days. Based on these data, it is proposed that thermotolerant PPR vaccines should maintain a minimum titre of 10^2.5^ TCID_50_/mL for vaccine dose on day 5 post-incubation at 40 °C with a titre loss below 1 log10 per mL. Preliminary immunogenicity test results showed that the PPR ThT vaccine meeting this criterion could be used in the field without maintaining a cold chain for up to 3 weeks, offering a practical solution for vaccination in remote areas.

## 1. Introduction

Peste des Petits Ruminants (PPR) is a highly contagious viral disease affecting small ruminants (sheep and goats) and wild small ruminants [1,2,3,4,5]. It is caused by PPR virus (PPRV) belonging to the *morbillivirus* genus within the family *Paramyxoviridae,* newly named small ruminant morbillivirus (SRMV), following the revision of virus nomenclature by the International Committee on Taxonomy of Viruses [6,7]. PPR virus (PPRV) exists as a single serotype and is classified into four distinct genetic lineages (I, II, III, and IV) [8,9,10]. Since the first description of PPR in Cote d’Ivoire in West Africa in 1942 [11], the disease has now been reported in over 70 countries in Africa, the Middle East, Asia and, recently, in some European countries (Georgia, Greece, Romania, Bulgaria, Hungary, and Albania) [12,13,14]. PPR seriously affects the livelihoods of small farmers in infected regions due to the mortality and morbidity rates, which can be as high as 50–90% and 10–90%, respectively, particularly in naive populations of sheep and goat [15]. Natural infection occurs in sheep and goats, and the fatality rate seems higher in goats than in sheep [16]. PPR virus can also infect wild ruminants [17] and has been causing severe epidemics in wild small ruminant populations in Asia, posing a threat to their conservation [1]. The disease is characterised by high pyrexia, necrosis and buccal erosion, pneumonia, leukopenia, diarrhoea, and shortness of breath [18]. Due to its significant impact on sheep and goats’ production and economic threats, the World Organisation for Animal Health (WOAH) classified Peste des Petits Ruminants (PPR) as a notifiable disease. With the successful global eradication of rinderpest and taking the lessons learnt from that project, the international veterinary community, with the support of the Food and Agriculture Organization (FAO) of the United Nations and WOAH, developed the PPR Global Control and Eradication Strategy (PPR GCES), adopted in 2015 in Abidjan, Cote d’Ivoire [19].

The control of PPR in most infected countries relies on vaccination using the two homologous PPR virus live attenuated vaccines based on the lineage II virus (Nigeria 75/1) [20] and the lineage IV virus (Sungri/96) [21,22]. The PPR Nigeria 75/1 vaccine is the most widely produced and used (AU-PANVAC PPR vaccine quality control report) globally, whereas the Lineage IV Sungri 96 vaccine is mainly used in India. Both vaccines stimulate a strong, long-term immune response to protect sheep and goats throughout their life and production period. However, the effective use of PPR vaccines in the field in many infected countries in hot climate regions is facing challenges in maintaining the required vaccine titre. Like most live attenuated vaccines, the PPR vaccine quality is affected by high temperatures, which can accelerate the degradation processes. Delivery of PPR vaccines to the field during vaccination campaigns will require the maintenance of the cold chain (+2–8 degrees Centigrade) with appropriate equipment, which could constitute a serious obstacle in some infected countries. Also, in most infected countries, the difficulty of having a reliable and stable source of electricity with proper refrigeration facilities is also a key challenge to the success of vaccination campaigns. To overcome all these challenges, the availability and use of PPR thermotolerant vaccines are essential. According to the WOAH, thermotolerant vaccines “*describe the ability to retain protective immunogenicity after exposure to temperatures above the storage temperature required according to the manufacturer’s recommendations. Claims of thermotolerance must be supported by data*” [23]. Thermotolerance can be achieved through a variety of technologies, including the use of thermotolerant vaccine strains, improved formulation of stabilisers and lyophilisation processes. For the existing live attenuated PPR vaccine strains, several approaches have been developed to make the vaccine thermotolerant. These processes are mainly based on the formulation of stabilisers and lyophilisation processes [24,25,26,27]. In December 2017, the FAO and WOAH PPR-Secretariat, with the support of the Global Alliance for Livestock Veterinary Medicines (GALVmed), organised a workshop on PPR thermotolerant vaccines at the FAO Headquarters in Rome. As one of the outcomes of the workshop, the participants requested WOAH and FAO, in collaboration with AU-PANVAC and any other relevant institutes, to develop a guideline on thermotolerance criteria to be applied to any PPR vaccine. It would be recalled that, during the Rinderpest (RP) eradication campaign, a test for the characterisation of thermostable vaccine was developed and proposed as the minimum standard for evaluation of RP vaccines marketed as heat stable [28]. The World Health Organisation (WHO) has also set up standards for heat stability of freeze-dried vaccines such as Measles, Mumps, and Rubella vaccines [29]. AU-PANVAC, with financial support from the GIZ (Germany) and the African Union programme budget, undertook a study for the development of criteria for evaluation of PPR thermotolerant vaccines.

## 2. Materials and Methods

### 2.1. Vaccine Receipt, Handling, and Registration

AU-PANVAC has developed a guideline for vaccine manufacturers for shipping vaccines to its facilities for quality control tests. As a requirement, all vaccines submitted for quality control testing are shipped under cold chain conditions (+4 to +8 °C). Upon arrival at the Addis Ababa airport, the PPR vaccines, just like all other animal vaccines, were transported in a cold van with a temperature around +4 to +8 °C to the AU-PANVAC facility located 45 km away from Addis Ababa, Ethiopia, following the customs clearance processes and after physical pre-inspection at the airport. Upon arrival at the AU-PANVAC facility, vaccines were registered and stored at +4 °C until testing.

Thirty-seven (37) freeze-dried PPR vaccine batches produced with the lineage II virus (Nigeria 75/1) from ten (10) vaccine manufacturers from Africa, the Middle East, and Asia were used in this study for the thermotolerance study. Among these vaccines, eight (8) were submitted as ThT PPR vaccines for the study, and twenty-nine (29) batches of conventional PPR vaccines requiring the maintenance of a cold chain were also included. Analysis of the stabilisers used in the production of the 37 freeze-dried PPR vaccines indicates the use of eight (8) different formulations, such as skimmed milk (SMK); lactalbumin–sucrose (LS); lactalbumin–sucrose–sodium glutamate (LSG), sucrose-peptone (SP), Lactose–NZ Amine (LA), Trehalose (TT), Weybridge medium (WB), and one unnamed product (Stabiliser30: SS).

All the PPR vaccine batches were given specific reference numbers following registration and blind tested by AUPANVAC laboratory technicians for quality control and thermotolerance study.

### 2.2. Incubation of Vaccines at High Temperature

All the PPR 37 vaccine freeze-dried vials stored at +4 °C were incubated at 40 °C and tested at day 3 and day 5 post-incubation to evaluate the impact of temperature on the titre.

In addition, the incubation period for seven (7) of the eight PPR vaccines sent as ThT vaccines was extended to 14 days at 40 °C. These vaccines were also incubated at 45 °C over the same 14-day period.

The temperature of the incubators was monitored daily, and records were taken over the period of incubation.

Vaccine vials in duplicate were collected for virus titration.

### 2.3. Residual Moisture Testing

Residual humidity in freeze-dried vaccine cake can impact the stability of the vaccine. All the 37 PPR vaccines were therefore tested for residual moisture using the thermogravimetric method [30]. Three vials of each vaccine were used for the test. Briefly, the vials were cleaned with ethanol 70%, the seals and the caps were removed, and the specimen sample (vaccine freeze-dried cake) placed in the Mettler Toledo (Mettler-Toledo GmbH, Laboratory & Weighing Technologies, CH-8606 Greifensee, Switzerland), a halogen moisture analyser that begins by weighing and recording the weight of the sample, then drying it using a halogen radiator while an integrated balance continuously records the weight of the sample. The total weight loss during all the processes is interpreted as the moisture content. The average of the readings of three vaccine vials is calculated as the residual moisture content in the PPR vaccine batch.

### 2.4. Titration of PPR Vaccine Virus

The initial titre of each vaccine formulation was determined using two vials of the same batches stored at +4 °C in the AU-PANVAC facility. The average titre of two samples of each vaccine formulation is considered the initial titre of the vaccine batch. Following incubation at 40 °C for 3 and 5 days, two vials of each vaccine batch were collected and titrated on the same day. Similar to the initial titre calculation, the average titre of the two samples of each vaccine formulation is considered as the final titre of the heat-treated vaccine batch. For each vaccine vial, virus titration was performed in a 96-well plate using Vero cells (prepared with G-MEM containing 10% fetal Calf serum and 1% antibiotic-antimycotic solution) after reconstitution with 2 mL of uncomplemented GMEM medium. Ten-fold dilutions of the virus were made (10^−1^ up to 10^−6^), and ten replicates of each dilution were prepared. Plates were incubated at 37 °C in a humidified incubator with 5% CO_2_ for 10 days. The plates were examined under a microscope for the appearance of cytopathic effect (CPE). Virus titre (log10 _TCID50/mL_) was calculated using the Spearman–Karber method [31].

### 2.5. Vaccination of Goats with Vaccines Incubated at 40 °C and 45 °C

Thirty (30) goats of about 12 months old were purchased and screened for PPR antibodies using the H-PPR bELISA [32] upon arrival at the AU-PANVAC animal laboratory facility. The animals were re-tested 21 days following the quarantine, and seronegative goats were divided into groups of 3 animals for vaccination. For this study, one batch of the vaccine submitted as ThT, PPR vaccine 1-SKM ThT vials were stored at 4 °C, 40 °C, and 45 °C. The vaccine vials were collected at day 7, day 21, and day 28 post-incubation and stored temporarily in the fridge at +4 °C for the vaccination of goats. Three goats vaccinated with a non-heat-treated vaccine 1-SKM ThT (stored at 4 °C) were also included as a control. On the same day, all vaccine vials were reconstituted in saline buffer as per the recommended doses by the manufacturer, and 1 mL was injected into the goat. Blood samples were collected at day 0 (vaccination day) and day 28 to evaluate the seroconversion using the HPPR-bELISA [32] and PPR-cELISA [33] supplied by Innovative Diagnostics (France), detecting antibodies directed against the hemagglutinin and nucleoprotein proteins, respectively. The tests have been conducted according to the protocol of each ELISA stated by the manufacturers. For the PPR-cELISA, the percentage of inhibition (PI) of each sample is interpreted as positive for PI < 50%, doubtful for PI 50–60%, and negative for PI > 60%. For the HPPR-bELISA, PI is interpreted as positive: PI ≥ 35%; doubtful: 30–35%; and negative: PI ≤ 30%.

## 3. Results

### 3.1. Residual Moisture in Vaccine Vials

Data presented in Table 1 showed that all PPR vaccine vials tested had a moisture content of less than 3% in accordance with the standard requirement [34].

### 3.2. Accuracy of PPR Vaccine Titration

The coefficient of variability (CV) was determined at 2.53% as the accuracy of the PPR vaccine from the titre of the internal reference PPR vaccine used as control in each plate.

### 3.3. Titres of PPR Vaccines Before and After the Incubation Period

Table 1 and Figure 1 show the titres of the PPR vaccine batches before and after incubation at 40 °C. All PPR vaccine batches showed an initial titre above the minimum required vaccine dose 10^2.5^ TCID_50_/mL, as per the WOAH standard [34]. The highest and lowest titre before incubation were 10^4.55^ TCID_50_/mL and 10^2.65^ TCID_50_/mL, respectively:-On Day 3, data showed that 6/8 (75%) vaccines submitted as ThT PPR maintain titre above the minimum required vaccine dose of 10^2.5^ TCID_50_/mL, and one (37-LSG ThT) with a titre slightly below the standard. For the conventional PPR freeze-dried vaccines, 15/29 (51.72%) presented titres above the threshold of 10^2.5^ TCID_50_/mL.-On Day 5, a total of 17 batches maintained a titre above the reference titre of 10^2.5^ TCID_50_/mL. Among these vaccines, 4/8 (50%) are vaccines submitted as ThT PPR vaccine, while 13/29 (44.82%) are conventional freeze-dried PPR vaccines. All these vaccines (ThT and conventional vaccines) are formulated with stabilisers such as skimmed milk, Stabiliser30, lactalbumin–sucrose, Lactose–NZ Amine, Weybridge, or Trehalose.

The mean total titre loss of all PPR vaccines maintaining titres greater than the minimum required vaccine dose of 10^2.5^ TCID50/mL after incubation at 40 °C on day 3 and day 5 was 0.78 log10 and 0.99 log10, respectively (Table 2).

Additionally, the seven batches submitted as ThT PPR vaccines incubated up to 14 days at 40 °C and 45 °C were titred as described above, and data are presented in Table 3 and Table 4. The degradation of the vaccine’s titre is presented by the regression curves in Figure 2. Three PPR ThT vaccines (1-SKM ThT, 6-LS THT, and 7-SS THT) maintained the titre above 10^2.5^ TCID_50_/mL at a 14-day period of incubation at 40 °C. All three vaccines exhibit a degradation curve with R^2^ values from 0.876 to 0.966, indicating a consistent and significant decline of the titre. The 37-LSG ThT showed stable or resistant to degradation following the storage at 40 °C, with an R^2^ value of 0.5. This remains relatively stable, suggesting a resistance to degradation with a titre that remained slightly below the standard. For the incubation condition at 45 °C, two batches, the 1-SKM ThT and 7-SS ThT, have continued to maintain titre above the standard at day 14.

### 3.4. Assessment of the Seroconversion Using the PPR Vaccine Without Maintaining the Cold Chain

The 1-SKM vaccine batch, which seems to have a relative good thermotolerance, was used for the study to determine the duration of time to use such a vaccine without maintaining to induce seroconversion in animals. The titres of the PPR ThT vaccine batch 1-SKM after incubation at 40 °C were 3.70, 2.55, and 2.30 log10 _TCID50/mL_ at 7, 21, and 28 days, respectively, while at 45 °C, the titres were 2.95, 2.05, and 1.35 log10 _TCID50/mL_ for the same incubation period. The immunogenicity of the 1-SKM PPR ThT vaccine incubated at various temperatures (4 °C, 40 °C, and 45 °C) was evaluated following the vaccination of goats. Sera samples were collected on day 0 and day 28 post-vaccination to assess seroconversion using the two serological assays, PPR-cELISA and HPPR-bELISA. The ELISA test results are summarised in Table 5. All vaccinated goats with the vaccine stored at 40 °C for 28 days showed 100% seroconversion 4 weeks after vaccination. However, at 45 °C, only the vaccine stored for 21 days can induce 100% seroconversion in vaccinated goats. A significant drop was observed in the vaccine storage at 45 °C on day 28, suggesting a decrease in efficacy after prolonged exposure beyond day 21. All control vaccinated goats with vaccine stored at 4 °C showed 100% seroconversion, and the unvaccinated controls remained seronegative.

## 4. Discussion

The present study assessed the stability and immunogenicity of conventional freeze-dried PPR vaccines and thermotolerant (ThT) vaccines under different incubation conditions to simulate the challenges in the field where maintaining the cold chain for vaccination will be difficult. Thirty-seven (37) PPR vaccine batches produced using the lineage II virus (Nigeria 75/1) were used for this study, and eight (8) were submitted as TT PPR vaccines, while twenty-nine (29) vaccine batches were conventional freeze-dried PPR vaccines. Prior to conducting the PPR vaccine thermotolerant study, residual moisture levels of all vaccine batches were determined and found to be below 3% as per the requirements, which indicates a good lyophilisation process by vaccine manufacturers. The residual moisture is critical for the preservation of the vaccine, as it can affect vaccine stability and potency. The initial titres of all PPR vaccine batches determined exceeded the WOAH minimum vaccine dose requirement of 10^2.5^ TCID_50_/mL before starting the study. The PPR vaccines were then exposed to 40 °C for 3 and 5 days to evaluate the impact of high temperature on the potency. Following incubation at 40 °C, a significant proportion of the PPR vaccine batches, 56.7% (21/37) and 46% (17/37), maintain titres above the standard at day 3 and day 5, respectively, regardless of whether the vaccines are conventional or thermotolerant freeze-dried formulations. Data showed that 50% (4/8) of the vaccines submitted as ThT vaccine and 44.82% (13/29) of conventional freeze-dried PPR vaccines had maintained titre above the standard on day 5 of incubation at 40 °C. These vaccines seem to indicate moderate heat sensitivity and were formulated using stabilisers such as skimmed milk, lactalbumin–sucrose, Lactose-NZ Amine, Weybridge, and Trehalose, as previously reported [26,35,36]. Also, the temperature stress test up to 14 days post-incubation at 40 °C was evaluated with seven batches submitted as ThT PPR vaccines, among which three batches (1-SKM ThT, 6-LS THT, and 7-SS THT) maintained their titre above or equal to 10^2.5^ TCID_50_/mL. The analysis of the kinetics of these vaccines showed a significant decline in titre (degradation), as shown with the regression curves R^2^ values from 0.876 to 0.966. However, the batch 37-LSG THT incubated under the same conditions seems more resistant to degradation, with a titre loss of 0.25 log10 over the incubation period of 14 days at 40 °C, as shown by the regression curves’ R^2^ values of 0.5.

Thus, it has become essential to determine the maximum titre loss to be considered in the criteria, making PPR ThT vaccines suitable for use in tropical conditions where it is difficult to maintain the cold chain. The mean titre loss of the vaccine batches maintaining titre above the required standard at day 3 and day 5 after incubation at 40 °C was 0.78 log10 and 0.99 log10, respectively. These data suggest that the titre loss that occurred during the first 3 days of incubation seems high. This parameter of titre loss is an essential criterion to be included to evaluate whether the PPR vaccine meets the ThT criteria, as is carried out for other attenuated viral vaccines. During the rinderpest eradication campaign, the thermostable vaccine was expected to maintain the minimum titre per vaccine dose, and the titre loss should not exceed 1.6 log10 after 14 days of storage at 45 °C [28]. Also, as per the World Health Organisation (WHO) requirements for heat stability of freeze-dried Measles, Mumps, and Rubella vaccines are evaluated, two indices (criteria) are used for the stability evaluation. These vaccines’ titres should retain at least 3 log10/dose, and titre loss should be below 1 log10 after 7 days of incubation at 37 °C [31]. For the PPR vaccine, based on the data from this study, it is proposed that ThT vaccines should maintain a minimum titre of 2.5 log10 for vaccine dose, and titre loss should not exceed 1 log10 per mL after 5 days of incubation at 40 °C.

The efficacy of PPR thermotolerant vaccines without a cold chain in the field was evaluated with the ThT vaccine. Although the 37-LSG ThT vaccine demonstrated the best thermotolerance among all the vaccines tested, we were unable to use this vaccine for the seroconversion study due to the limited number of vials supplied by the manufacturer. The 1-SKM ThT vaccines showing a relative thermotolerance were used in this study, as evidenced by a titre loss of 0.6 log10 between day 0 and day 5, compared to the mean titre loss of 1 log10 among vaccines that retained titres greater than 2.5 log10 on day 5 of incubation at 40 °C. The 1-SMK vaccine vials stored at 4 °C and 40 °C for 28 days induced 100% seroconversion in goats by day 28 post-vaccination, indicating that the vaccine preserved its immunogenicity. At 45 °C, only vials from 21 days post-incubation induced 100% seroconversion in vaccinated animals, suggesting that prolonged exposure to high temperatures eventually compromises the vaccine’s efficacy, as it can be observed with the drop of vaccine titre to 1.35 log10 _TCID50/mL_. These preliminary data need to be evaluated in the field conditions.

## 5. Conclusions

The data from this study strongly supports the use of two parameters as proposed criteria for the evaluation of thermotolerant PPR vaccine after incubation at 40 °C: (1) to maintain the minimum titre of 10^2.5^ TCID_50_/mL for vaccine dose, and (2) the titre loss must not exceed 1 log10 after 5 days incubation. The preliminary data shows that PPR vaccines meeting such criteria can maintain their immunogenicity for up to 21 days without a cold chain. The use of such ThT vaccines in the field for PPR vaccination programmes could strongly contribute to supporting the global eradication of PPR, particularly in tropical and sub-tropical regions where it is difficult to maintain the cold chain.

## Figures and Tables

**Figure 1 viruses-17-01151-f001:**
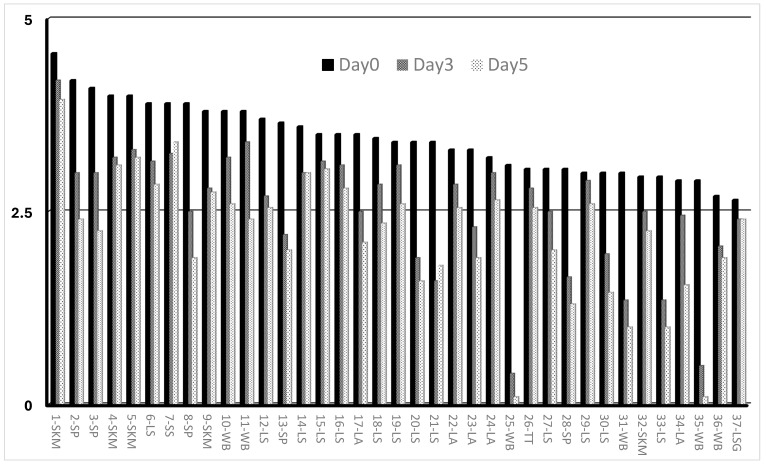
Histogram chart of PPR vaccines titres following incubation at 40 °C after day 3 and day 5. The *X*-axis represents the different vaccines, while the *Y*-axis shows the log10 of vaccine titre after incubation at 40 °C. All the vaccine batches at incubation time with a titre above the 2.5 log10 _TCID50/mL_ meet the reference standard titre of the PPR vaccine.

**Figure 2 viruses-17-01151-f002:**
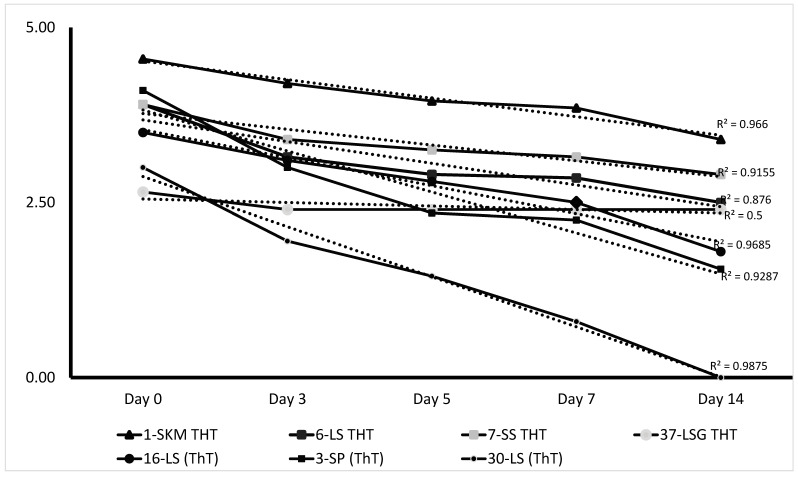
Degradation curve of PPR vaccine batches over the 14-day incubation at 40 °C. The markers represent the actual data points of each vaccine at the incubation period, and the dotted lines show the fitted linear regression models for each sample. The *X*-axis represents the dates of incubation, while the *Y*-axis shows the log10 of the vaccine’s titre after incubation at 40 °C.

**Table 1 viruses-17-01151-t001:** Titres of the PPR vaccine batches following the incubation at 40 °C on day 3 and day 5. Shaded data represent vaccine batches maintaining titre above the 2.5 log10 _TCID50/mL_.

		Vaccine Titre (log10 _TCID50/mL_)			Titre Loss (log10 _TCID50/mL_)	
	Residual moisture [≤3%]	Day 0	Day 3	Day 5	Day 0–Day 3	Day 0–Day 5
1-SKM (ThT)	1.72%	4.55	4.2	3.95	0.35	0.6
2-SP	2.0%	4.2	3	2.4	1.2	1.8
3-SP (ThT)	1.52%	4.1	3	2.25	1.1	1.85
4-SKM	1.7%	4	3.2	3.1	0.8	0.9
5-SKM	1.1%	4	3.3	3.2	0.7	0.8
6-LS (ThT)	1.17%	3.9	3.15	2.90	0.75	1
7-SS (ThT)	0.76%	3.9	3.4	3.25	0.5	0.65
8-SP	2.1%	3.9	2.5	1.9	1.4	2
9-SKM	1.26%	3.8	2.8	2.75	1	1.05
10-WB	2.3%	3.8	3.2	2.6	0.6	1.2
11-WB	3.0%	3.8	3.4	2.4	0.4	1.4
12-LS	2.51%	3.7	2.7	2.55	1	1.15
13-SP	1.9%	3.65	2.2	2	1.45	1.65
14-LS	1.1%	3.6	3	3	0.6	0.6
15-LS	1.1%	3.5	3.15	3.05	0.35	0.45
16-LS (ThT)	0.88%	3.5	3.1	2.8	0.4	0.7
17-LA	1.6%	3.5	2.5	2.1	1	1.4
18-LS (ThT)	0.84%	3.45	2.85	2.35	0.6	1.1
19-LS	1.6%	3.4	3.1	2.6	0.3	0.8
20-LS	2.9%	3.4	1.9	1.6	1.5	1.8
21-LS	2.2%	3.4	1.6	1.8	1.8	1.6
22-LA	1.7%	3.3	2.85	2.55	0.45	0.75
23-LA	1.9%	3.3	2.3	1.9	1	1.4
24-LA	1.8%	3.2	3	2.65	0.2	0.55
25-WB	2.6%	3.1	0.4	0.1	2.7	3
26-TT	2.2%	3.05	2.8	2.55	0.25	0.5
27-LS	2.5%	3.05	2.5	2	0.55	1.05
28-SP	2.28%	3.05	1.65	1.3	1.4	1.75
29-LS	2.3%	3	2.9	2.6	0.1	0.4
30-LS (ThT)	2.17%	3	1.95	1.45	1.05	1.55
31-WB	3.8%	3	1.35	1	1.65	2
32-SKM	2.2%	2.95	2.5	2.25	0.45	0.7
33-LS	2.8%	2.95	1.35	1	1.6	1.95
34-LA	2.7%	2.9	2.45	1.55	0.45	1.35
35-WB	2.8%	2.9	0.5	0.1	2.4	2.8
36-WB	2.4%	2.7	2.05	1.9	0.65	0.8
37-LSG (ThT)	n,d	2.65	2.4	2.4	0.25	0.25

**Table 2 viruses-17-01151-t002:** The mean titre loss of PPR vaccines with a titre greater than the required 10^2.5^ TCID50/mL after incubation at 40 °C on Day 3 and Day 5.

	Day 0–Day 3	Day 0–Day 5	Day 3–Day 5
Average Titre Loss	0.52 log10	0.72 log10	0.20 log10
SD	0.26	0.27	0.2
Average Titre Loss + SD	0.78 log10	0.99 log10	0.40 log10

**Table 3 viruses-17-01151-t003:** Titres (log10) of the PPR vaccine batches submitted as thermotolerant vaccine after incubation at 40 °C and 45 °C for 14 days.

	1-SKM THT		3-SP (ThT)		6-LS THT		7-SS THT		16-LS (ThT)		30-LS (ThT)		37-LSG THT	
Incubation Time	40 °C	45 °C	40 °C	45 °C	40 °C	45 °C	40 °C	45 °C	40 °C	45 °C	40 °C	45 °C	40 °C	45 °C
Day 0	4.55	4.55	4.10	4.10	3.90	3.9	3.90	3.9	3.50	3.50	3.00	3.00	2.65	2.65
Day 3	4.20	3.95	3.00	2.00	3.15	3	3.40	3.25	3.10	2.80	1.95	0	2.40	2.4
Day 5	3.95	3.65	2.35	1.95	2.90	2	3.25	3	2.80	2.45	1.45	0	2.40	2.4
Day 7	3.85	3.1	2.25	1.80	2.85	1.85	3.15	2.85	2.50	2.05	0.80	0	2.40	2.4
Day 14	3.40	2.7	1.55	1.50	2.50	1.5	2.90	2.55	1.80	1.15	0.00	0	2.40	1.9

**Table 4 viruses-17-01151-t004:** Titre loss of PPR vaccine batches submitted as thermotolerant vaccine after incubation at 40 °C and 45 °C for 14 days.

	1-SKM THT		3-SP (ThT)		6-LS THT		7-SS THT		16-LS (ThT)		30-LS (ThT)		37-LSG THT	
	40 °C	45 °C	40 °C	45 °C	40 °C	45 °C	40 °C	45 °C	40 °C	45 °C	40 °C	45 °C	40 °C	45 °C
Days 0–3	0.35	0.6	1.1	2.1	0.75	0.9	0.5	0.65	0.4	0.7	1.05	3	0.25	0.25
Days 0–5	0.6	0.9	1.75	2.15	1	1.9	0.65	0.9	0.7	1.05	1.55	3	0.25	0.25
Days 0–7	0.7	1.45	1.85	2.3	1.05	2.05	0.75	1.05	1	1.45	2.2	3	0.25	0.25
Days 0–14	1.15	1.85	2.55	2.6	1.4	2.4	1	1.35	1.7	2.35	3	3	0.25	0.75

**Table 5 viruses-17-01151-t005:** Vaccinated goats with PPR vaccine (1-SKM PPR ThT) incubated at 40 °C and 45 °C on day 7, day 21, and day 28. Seroconversion detection with ELISAs (PPR-cELISA and HPPR-bELISA) on sera collected at day 0 and day 28 post-vaccination.

Incubation of 1-SKM PPR ThT Vaccine		Number of Vaccinated Goats	ELISAs Results: (%) POSITIVE			
Temperature	Incubation Time		IDVET PPR cELISA		HPPR bELISA	
			Sera Collected at Day 0	Sera Collected at Day 28	Sera Collected at Day 0	Sera Collected at Day 28
+4 °C	Day 0	3	0	3/3 (100%)	0	3/3 (100%)
40 °C	Day 7 (titre of 3.70 log10 _TCID50/mL_)	3	0	3/3 (100%)	0	3/3 (100%)
	Day 21 (titre 2.55 log10 _TCID50/mL_)	3	0	3/3 (100%)	0	3/3 (100%)
	Day 28 (titre 2.30 log10 _TCID50/mL_)	3	0	3/3 (100%)	0	3/3 (100%)
45 °C	Day 7 (titre 2.95 log10 _TCID50/mL_)	3	0	3/3(100%)	0	3/3 (100%)
	Day 21 (titre 2.05 log10 _TCID50/mL_)	3	0	3/3 (100%)	0	3/3 (100%)
	Day 28 (titre 1.35 log10 _TCID50/mL_)	3	0	2 (66.66%)	0	1/3 (33%)
Unvaccinated		2	0	0	0	0

## Data Availability

The data are contained in the article. Also, additional information can be obtained upon request from the corresponding author.

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
