# Peer review of "Peste des Petits Ruminants Vaccine: Criteria for Assessing Its Thermotolerance"

_viruses, 2025, doi:10.3390/v17091151_

Round 1

Reviewer 1 Report

Comments and Suggestions for Authors

1 title, add proposed before criteria.

2 2.4, last sentence, TCID50/ml should be subscript.

3 Table 1, % was missing for some of the data.

4 Table 1, what does the shaded content mean?

5 Table 3, supplement titer loss.

6 3.1, the reference standard should be clarified here.

7 Table 1 and Table 3, did the author test the titer of some of the non-ThT vaccines at 7 and 14 days? According to Table 1, some of the non-ThT vaccines, such as 5SKM, 14 LS, 15LS, also had lower titer loss, which is even better than some of the ThT vaccine. 

8 Table 4 showed the results of vaccinated with 1-SKM incubated at 40 and 45 °C for 21 and 28 days, but the titers at these time points were not shown in table 3 or other places in the text.

9 3.4, based on the results presented in the manuscript, the 1-SKM vaccine was not the best ThT vaccine, just because the starting titer is the highest. The titer loss of 1-SKM was not the smallest. 37-LSG THT seems to be the best ThT vaccine, with a titer loss of only 0.25.

10 For ThT vaccines, the titer of >102.5 TCID50/ml on Day5 post-incubation at 40°C is not the key indicator, the titer loss is the crucial one. The titer of >102.5 TCID50/ml is the basic indicator to ensure the effectiveness of all PPR vaccine, no matter what kind it is (ThT or non-ThT). This should be clarified in the text.

11 Table 4, The number of immunized animals (n=3) is insufficient.

12 Table 1 and Table 4, what’s the * mean?

Author Response

Responses to Reviewers 1: Comments and Suggestions for Authors

  • We thank this reviewer for his valuable comments, and we have addressed them in rebuttals as well as in the text/table of the manuscript as required.
  1. title, add proposed before criteria.
  • Response: Accepted and editing done
  1. 4, last sentence, TCID50/ml should be subscript.
  • Response: Accepted and editing done
  1. Table 1, % was missing for some of the data.
  • Response: Accepted and editing done
  1. Table 1, what does the shaded content mean?
  • Response: Shaded data represent vaccine batches maintaining a titer above the 2.5 log10 CID50/ml. This was highlighted in the text.
  1. Table 3, supplement titer loss.
  • Response: Accepted and editing done
  1. 1, the reference standard should be clarified here.
  • Response: Accepted and editing done (Table 3.2)
  1. Table 1 and Table 3, did the author test the titer of some of the non-ThT vaccines at 7 and 14 days? According to Table 1, some of the non-ThT vaccines, such as 5SKM, 14 LS, 15LS, also had lower titer loss, which is even better than some of the ThT vaccine. 
  • Response:
    • PPR vaccines named ThT were submitted by manufacturers with enough vials for the purpose of the study. Non-ThT vaccines were tested for comparison and the remaining vials, which were submitted for routine quality control by manufacturers, were not enough to conduct the test at 7 and 14 days.
    • Yes, some of the non-ThT vaccines, had lower titer loss, better than some of the ThT vaccine. This was highlighted in the text.
  1. Table 4 showed the results of vaccinated with 1-SKM incubated at 40 and 45 °C for 21 and 28 days, but the titers at these time points were not shown in table 3 or other places in the text.
  • Response: The titers of the PPR ThT vaccine batch 1-SKM after incubation at 40°C were 2.55 and 2.30 log10 at 21 and 28 days, respectively, while at 45°C, the titers were 2.05 and 1.35 log10 for the same incubation period. Edited in track changed mode in chapter 3.4.

Titre: PPR TT vaccine Batch P035

Titer at Day 0: 104.35 TCID(50)/mL

Interval

40°C

45°C

Day 7

3.75

2.95

Day 21

2.55

2.05

Day 28

2.3

1.35

  1. 4, based on the results presented in the manuscript, the 1-SKM vaccine was not the best ThT vaccine, just because the starting titer is the highest. The titer loss of 1-SKM was not the smallest. 37-LSG THT seems to be the best ThT vaccine, with a titer loss of only 0.25.
  • Response:
  • 1-SMK which seems simulate good ThT criteria as indicated titer loss of 0.6 log10 between Day 0 and day 5 compared to the mean of titer loss (1 log10) from those vaccines maintain titer above the 2.5 log10 at day 5 incubation at 40°C. Yes, 37-LSG THT compared to 1-SMK, seems to be the best ThT vaccine, with a titer loss of only 0.25.
  • We did not use the vaccine 37-LSG THT to assess the Sero-Conversion due to the limited number of vials submitted. We contacted the manufacturer, but they were unable to provide additional vials of this batch. When 1-SMK vaccine formulation could generate 100% seroconversion, there is no doubt that 37-LSG THT could have done similar or better.
  1. For ThT vaccines, the titer of >105TCID50/ml on Day5 post-incubation at 40°C is not the key indicator, the titer loss is the crucial one. The titer of >102.5TCID50/ml is the basic indicator to ensure the effectiveness of all PPR vaccine, no matter what kind it is (ThT or non-ThT). This should be clarified in the text.
  • Response: Comment accepted. That is why it is proposed to use the dual parameters of standard titer of >105TCID50/ml on Day5 post-incubation at 40°C and titer loss must not exceed of 101 TCID50/ml to assess PPR ThT vaccine.
  1. Table 4, The number of immunized animals (n=3) is insufficient.
  • Response: The comment concerning the number of immunised animals is noted; however, including more number of animals could be useful butdata from the presentthe study can provides valuable information on seroconversion of vaccinated animals with vaccine incubated at 40°C over the specified period of 21 days.
  1. Table 1 and Table 4, what’s the mean?
  • Response: The asterisk * was used to identify data just below the reference. It has been deleted from the text.

Reviewer 2 Report

Comments and Suggestions for Authors

 PPR live attenuated vaccines are thermolabile and their use requires maintaining the cold chain from the manufactory premises to the field. To address this challenge, the authors focused on developing thermotolerant (ThT) PPR vaccines using different stabilisers and improving the freeze-drying process. There were some main concerns.

  1. Thirty-seven (37) freeze-dried PPR vaccine batcheswere used in the text, however, in each group, there were only one batch. Each group of experiments requires at least three biological replicates. The experiments in this article did not meet the requirements for statistical analysis.
  2. Ref (Veterinary sciences, 2024, 11, 525. 189https://doi.org/10.3390/vetsci11110525) have evaluated the thermotolerance of various for mulations of PPR vaccines, Did the author compare the differences in the heat stabilizer effects between the two ThT PPR vaccines ?

Author Response

Responses to Reviewer 2: Comments and Suggestions for Authors

  • We thank this reviewer for his valuable comments, and we have addressed them in rebuttals as well as in the text/table of the manuscript as required.

 PPR live attenuated vaccines are thermolabile and their use requires maintaining the cold chain from the manufactory premises to the field. To address this challenge, the authors focused on developing thermotolerant (ThT) PPR vaccines using different stabilisers and improving the freeze-drying process. There were some main concerns.

  1. Thirty-seven (37) freeze-dried PPR vaccine batches were used in the text, however, in each group, there were only one batch. Each group of experiments requires at least three biological replicates. The experiments in this article did not meet the requirements for statistical analysis.
  • The reviewer comment is noted; this is a preliminary study; however, the authors think that the study provides valuable information on the preliminary criteria to use to assess PPR ThT vaccines. The 37 vaccines batches were collected from ten (10) vaccine manufacturers from Africa, the Middle East, and Asia. Also, based on our database, a limited number of PPR vaccine manufacturers are developing a programme for ThT vaccine production.
  1. Ref (Veterinary sciences, 2024, 11, 525. 189 https://doi.org/10.3390/vetsci11110525) have evaluated the thermotolerance of various formulations of PPR vaccines, Did the author compare the differences in the heat stabilizer effects between the two ThT PPR vaccines?
  • Manufactuerers are using stablizers such as Skim milk (SMK); Lactalbumin-Sucrose (LS); Lactalbumin-sucrose-sodium glutamate (LSG), Sucrose-Peptone (SP), Lactose - N-Z Amine (LA), Trehalose (TT), Weybridge medium (WB) and one unnamed product (Stabilizer30: SS) for PPR vaccine formulation for ThT or non-ThT.
  • This study was as the continuation of the previous reported data indicating that vaccines formulated using stabilizers such as Skimmed milk; Lactalbumin-Sucrose; Lactose- N-Z Amine; Weybridge and Trehalose shown moderate heat sensitivity [Veterinary sciences, 2024, 11, 525. 189 https://doi.org/10.3390/vetsci11110525]. This reference in mentioned in the text as [35].

Reviewer 3 Report

Comments and Suggestions for Authors

The paper is well written and the data is well presented.

My main concern is the conclusions are not well supported by the methods and data.

The minimum titer standard of 2.5 log 10 is well established in international norms and proven through years of use. If a vaccine retains 2.5 logs 10 at the time of use in the field it will lead to an effective immune response. The suggestion of no more than 1 log loss in titer as an additional standard is not supported by the data presented, in vitro or in vivo. It is not related to any analysis of the kinetics and nature of the vaccines degradation. 

Although field conditions can be quite harsh, the average daily temperature in the most severe pastoral areas does not exceed about 28-29C. In the afternoon, temperatures can reach the 45 to 50C range. It is the vaccines ability to tolerate accumulated temperature stress and temperature shocks that is relevant. At the same time, temperatures in confined spaces such as cars or market sheds can reach higher levels. Rather than arbitrarily shorten the storage life outside of the cold chain from 28 to 21 days, it would be more appropriate to define appropriate storage conditions as is done for other medications that do not require refrigeration and define a storage life based recommended handling with a reasonable margin of safety.

Author Response

Responses to Reviewer 3: Comments and Suggestions for Authors

The paper is well written, and the data is well presented.

  • We thank this reviewer for his valuable comments, and we have addressed them in rebuttals as well as in the text/table of the manuscript as required.

My main concern is the conclusions are not well supported by the methods and data.

The minimum titer standard of 2.5 log 10 is well established in international norms and proven through years of use. If a vaccine retains 2.5 logs 10 at the time of use in the field it will lead to an effective immune response. The suggestion of no more than 1 log loss in titer as an additional standard is not supported by the data presented, in vitro or in vivo. It is not related to any analysis of the kinetics and nature of the vaccines degradation. 

  • Response:
  • The authors agreed that all PPR vaccines should meet the minimum titer standard of 2.5 log 10 for effective immune response for protection of animals. However, if we want to guarantee ThT PPR vaccine (which is supposed to be transported from the storage up the field) to meet this standard, integration of the titer loss as one parameter during the vaccine quality control should be taking.
  • During the rinderpest eradication campaign, two parameters were used to assess the thermostable vaccine by maintain the minimum titer per vaccine dose and the titer loss should not exceed 1.6 log10 after 14 days storage at 45°C. Similarly, the World Health Organization (WHO) proposed for heat stability of freeze-dried Measles, Mumps and Rubella vaccines to use the 2 indices of minimum titer standard of 3 log10 /dose and titer loss must not exceed 1 log10 after 7 days of incubation at 37°C.
  • The analysis of the kinetics of most vaccines incubated at 40°C up to day 14 showed a significant decline of titer (degradation), except for the vaccine formulated with 37-LSG THT, and this is shown in a regression curve R² values from 0.876 to 0.966.

Although field conditions can be quite harsh, the average daily temperature in the most severe pastoral areas does not exceed about 28-29C. In the afternoon, temperatures can reach the 45 to 50C range. It is the vaccines’ ability to tolerate accumulated temperature stress and temperature shocks that is relevant. At the same time, temperatures in confined spaces such as cars or market sheds can reach higher levels. Rather than arbitrarily shorten the storage life outside of the cold chain from 28 to 21 days, it would be more appropriate to define appropriate storage conditions as is done for other medications that do not require refrigeration and define a storage life based recommended handling with a reasonable margin of safety.

  • Response: The study has been conducted using a simulated field temperature at 40°C as an average, knowing that in some areas the temperature could be higher. The storage life of 21 days at ambient temperature was determined based on titers of vaccines incubated at 40°C and 45°C. On 14 days post-incubation, the vaccines 1-SKM THT, 6-LS THT and 7-SS THT retained titer above the standard for temperature stress at 40°C, while only 1-SKM THT and 7-SS THT retained titer at 45°C. In addition, the vaccine 1-SKM THT incubated at 40°C and 45°C failed below the standard on 28 days and Day 21.

Reviewer 4 Report

Comments and Suggestions for Authors

The authors evaluate PPR vaccines with respect to assessing criteria for thermotolerance. This is of importance due to the lack of cold chains in many regions where PPR is present and having PPR vaccines that can be used in the absence of a cold chain can facilitate the use of these vaccines in the field.

The authors have done detailed virus titration following heating at different timepoints 3 and 5 days.

Minor corrections

in the abstract Day5 post-incubation replace with day 5

Introduction line 3 Paramyxoviridae, newly fix the font size.

For the serology of the vaccinated goats 2 different ELISAs were used. It would be nice to see VNT data including the titres to show the efficacy of the vaccines tested. This would improve the paper to include this data.

Discussion line 28 residual

Author Response

Response to Reviewer 4: Comments and Suggestions for Authors

The authors evaluate PPR vaccines with respect to assessing criteria for thermotolerance. This is of importance due to the lack of cold chains in many regions where PPR is present and having PPR vaccines that can be used in the absence of a cold chain can facilitate the use of these vaccines in the field.

The authors have done detailed virus titration following heating at different timepoints 3 and 5 days.

  • We thank this reviewer for his valuable comments, and we have addressed them in rebuttals as well as in the text/table of the manuscript as required.

Minor corrections

  • Response: The authors thank the reviewer for his kind comment

in the abstract Day5 post-incubation replace with day 5

  • Response: accepted, editing done.

Introduction line 3 Paramyxoviridae, newly fix the font size.

  • Response: accepted, editing done.

For the serology of the vaccinated goats 2 different ELISAs were used. It would be nice to see VNT data including the titres to show the efficacy of the vaccines tested. This would improve the paper to include this data.

  • Response: We accepted the comment, however the HPPR-bELISA is directed to detect neutralizing antibodies against the PPRV H protein. This assay has shown a correlation of Kappa agreement 0.947 with VNT (Arch Virol., 2018, 163:1745-1756. https://doi.org/10.1007/s00705-018-3782-1)

Discussion line 28 residual

  • Response: accepted, editing done.

Round 2

Reviewer 1 Report

Comments and Suggestions for Authors

1. 2.4, last sentence, TCID50/ml, 50 should be subscript!!!

2. Table 1 title, TCID50/ml, T is missing and 50 should be subscript!

3. 3.4, line 5-7, add the titers of different time points (0, 7, 21, 28) in table 4.

4. The shaded vaccines in table 1 should also be marked with a symbol at the x-axis in Figure 1. 

5. Table 2 could be deleted. The mean titer loss of all PPR vaccines is meaningless for this study.

6. Based on the results presented in the manuscript, the 1-SKM vaccine was not the best ThT vaccine, the titer loss of 1-SKM was high. While, 37-LSG THT seems to be the best ThT vaccine, with a titer loss of only 0.25, suggesting that the stabilisers or lyophilisation process is the best. The improvement of stabilisers or lyophilisation process of ThT vaccines and the promote of the technology used for 37-LSG THT (and 14 LS, 15LS, 29LS, etc) is important for PPRV ThT vaccines development. This might be clarified further in the disscussion part.

Author Response

The authors would like to thank the reviewer for his valuable comments, and we have addressed them in rebuttals as well as in the text/table of the manuscript as required.

  1. 4, last sentence, TCID50/ml, 50 should be subscript!!!

Response: Accepted and editing done

  1. Table 1 title, TCID50/ml, T is missing and 50 should be subscript!

Response: Accepted and editing done

  1. 4, line 5-7, add the titers of different time points (0, 7, 21, 28) in table 4.

Response: Accepted and editing done

  1. The shaded vaccines in table 1 should also be marked with a symbol at the x-axis in Figure 1. 

Response: Accepted and editing done

The sentence “All the vaccine batches at incubation time with titer above the 2.5 log10TCID50/ml meet the reference standard titer of PPR vaccine” added in the description of Figure 1.

  1. Table 2 could be deleted. The mean titer loss of all PPR vaccines is meaningless for this study.

Response:

The mean titer loss was ONLY calculated for PPR vaccine batches that retained titers above the required 102.5 TCID50/mL following incubation at 40°C on Day 3 (21 out of 37) and on Day 5 (17 of the 37).

The authors believe that this provides valuable information on the dynamics of the titer degradation under temperature stress of the different formulation of PPR vaccines.

  1. Based on the results presented in the manuscript, the 1-SKM vaccine was not the best ThT vaccine, the titer loss of 1-SKM was high. While, 37-LSG THT seems to be the best ThT vaccine, with a titer loss of only 0.25, suggesting that the stabilisers or lyophilisation process is the best. The improvement of stabilisers or lyophilisation process of ThT vaccines and the promote of the technology used for 37-LSG THT (and 14 LS, 15LS, 29LS, etc) is important for PPRV ThT vaccines development. This might be clarified further in the disscussion part.

Response: Accepted and editing done. This was reiterated in the text, 4. Discussion (Lines: 47 – 51 and 70 – 71)

With Kind regards

Reviewer 2 Report

Comments and Suggestions for Authors

Each group of experiments requires at least three biological replicates. The experiments in this article did not meet the requirements for statistical analysis. 

In fig.1, table 2, table 3.1 and 3.2, each test has only one repetition. The credibility of the data is in question.

Author Response

The authors would like to thank the reviewer for his valuable comments, and we have addressed them in rebuttals as well as in the text/table of the manuscript as required.

Each group of experiments requires at least three biological replicates. The experiments in this article did not meet the requirements for statistical analysis. 

In fig.1, table 2, table 3.1 and 3.2, each test has only one repetition. The credibility of the data is in question.

Response:

The reviewer comment is noted.

The authors would like to indicate that 37 vaccines batches collected from ten (10) vaccine manufacturers were tested for this study.

As indicated in the material and methods, two vials of each vaccine batch titrated a testing point on Day 0 (+4°C), Day 3 and 5 (incubation at 40°C). Vaccines submitted as ThT were also tested for on day 7 and Day 14 at 40°C and at 45°C. The titration of the dilutions (10-1 up to 10-6) of each vaccine vial was conducted in 10 replicates in the plates.

In addition, during each vaccine batch titration, an internal reference PPR vaccine was used as control and the coefficient of variability (CV) from the titer of this vaccine was calculated at 2.53% indicating the accuracy of the PPR vaccine titration performed.

At least 500 titrations of PPR vaccines in microplates were performed.

With kind regards
